# Patterns of recent natural selection on genetic loci associated with sexually differentiated human body size and shape phenotypes

Audrey M. Arner[1]*, Kathleen E. Grogan[1,2], Mark Grabowski[3,4], Hugo Reyes-Centeno[3,5], George H. Perry[1,3,6]*

1 Departments of Anthropology and Biology, Pennsylvania State University, University Park, Pennsylvania, United States of America, 2 Departments of Anthropology and Biology, University of Cincinnati, Cincinnati, Ohio, United States of America, 3 DFG Center for Advanced Studies, University of Tübingen, Tübingen, Germany, 4 Research Centre in Evolutionary Anthropology and Palaeoecology, Liverpool John Moores University, Liverpool, United Kingdom, 5 Department of Anthropology & William S. Webb Museum of Anthropology, University of Kentucky, Lexington, Kentucky, United States of America, 6 Huck Institutes of the Life Sciences, Pennsylvania State University, University Park, Pennsylvania, United States of America

* arneraudreym@gmail.com (AMA); ghp3@psu.edu (GHP)

**Data Availability Statement:** All data files are available in the Dryad Digital Repository: https://doi.org/10.5061/dryad.nzs7h44rc.

## Abstract

Levels of sex differences for human body size and shape phenotypes are hypothesized to have adaptively reduced following the agricultural transition as part of an evolutionary response to relatively more equal divisions of labor and new technology adoption. In this study, we tested this hypothesis by studying genetic variants associated with five sexually differentiated human phenotypes: height, body mass, hip circumference, body fat percentage, and waist circumference. We first analyzed genome-wide association (GWAS) results for UK Biobank individuals (~194,000 females and ~167,000 males) to identify a total of 114,199 single nucleotide polymorphisms (SNPs) significantly associated with at least one of the studied phenotypes in females, males, or both sexes (P<5x10^-8). From these loci we then identified 3,016 SNPs (2.6%) with significant differences in the strength of association between the female- and male-specific GWAS results at a low false-discovery rate (FDR<0.001). Genes with known roles in sexual differentiation are significantly enriched for co-localization with one or more of these SNPs versus SNPs associated with the phenotypes generally but not with sex differences (2.73-fold enrichment; permutation test; P = 0.0041). We also confirmed that the identified variants are disproportionately associated with greater phenotype effect sizes in the sex with the stronger association value. We then used the singleton density score statistic, which quantifies recent (within the last ~3,000 years; post-agriculture adoption in Britain) changes in the frequencies of alleles underlying polygenic traits, to identify a signature of recent positive selection on alleles associated with greater body fat percentage in females (permutation test; P = 0.0038; FDR = 0.0380), directionally opposite to that predicted by the sex differences reduction hypothesis. Otherwise, we found no evidence of positive selection for sex difference-associated alleles for any other trait. Overall, our results challenge the longstanding hypothesis that sex differences adaptively decreased following subsistence transitions from hunting and gathering to agriculture.

**Funding:** This work was funded by the National Institutes of Health (NIH) grant R01-GM115656 (to G.H.P.); NIH grant F32-GM123634 (to K.E.G.); Deutsche Forschungsgemeinschaft (DFG) grant FOR-22337 (to A.M.A, M.G, H.R.C, and G.H.P.); and Erickson Discovery (https://urfm.psu.edu/research/erickson-discovery-grant), Presidential Leadership Academy Enrichment (https://academy.psu.edu/current/grants/), and Liberal Arts Enrichment grants (https://la.psu.edu/current-students/undergraduate-students/scholarships-and-funding/enrichment-funding) from Penn State University (to A.M.A). The funders had no role in study design, data collection and analysis, decision to publish, or preparation of the manuscript.

**Competing interests:** The authors have declared that no competing interests exist.

## Author summary

There is uncertainty regarding the evolutionary history of human sex differences for quantitative body size and shape phenotypes. In this study we identified thousands of genetic loci that differentially impact body size and shape trait variation between females and males using a large sample of UK Biobank individuals. After confirming the biological plausibility of these loci, we used a population genomics approach to study the recent (over the past ~3,000 years) evolutionary histories of these loci in this population. We observed significant increases in the frequencies of alleles associated with greater body fat percentage in females. This result is contradictory to longstanding hypotheses that sex differences have adaptively decreased following subsistence transitions from hunting and gathering to agriculture.

## Introduction

In many vertebrate species, it is not uncommon for morphological phenotypes to have average size and shape differences between females and males [1]. Traits with average phenotype values that differ by sex but with overlapping trait distributions–such as human height and body fat percentage–are described as 'sexually differentiated' traits. Technically, the more commonly-used term 'sexually dimorphic' is specific to non-overlapping traits; for example, exaggerated ornamentation in male guppies, peacocks, and mandrills [2–5].

Some sexually differentiated traits are believed by many researchers to be the result of sexual selection. In species with high inter-male competition for mates, larger males may have a competitive advantage that results in increased fitness [6]. Perhaps as a result, the magnitude of sexually differentiated phenotypes are often greater in polygynous species with high competition (e.g. gorillas) and lower in monogamous species (e.g. gibbons) [6–8]. Finally, there are major differences in the degree of body size and shape sex differences between closely related species, suggesting the potential for the relatively rapid evolution of sexually differentiated traits [6].

In humans, females and males exhibit significant but relatively subtle differences in many anthropometric phenotypes [9]. For example, European and African males are an average of approximately 9% taller in height and 15% heavier in body mass than females from the same populations [10,11]. Humans also exhibit sexually differentiated biometric and disease phenotypes, especially those related to immune function [12,13].

There is little consensus regarding the evolutionary history of sexually differentiated traits in our species [11,14,15]. Current levels of human sex differences may partially reflect an evolutionary history of complex interactions between our biology and culture, but the timing, direction, and forces responsible for any adaptive changes in these patterns are debated. For example, it has been hypothesized that the degree of sexual differentiation for body size phenotypes in Europe likely decreased following recent (within the past ~10,000 years) shifts from a foraging-based subsistence strategy to one relying primarily on food production, in response to less pronounced divisions of labor and mobility [16]. However, other scholars suggest that sexually differentiated phenotypes would respond to selection at too slow an evolutionary rate to respond to recent environmental and cultural changes; thus, any such recent changes are more likely to reflect genetic drift and/or non-genetic responses to environmental changes rather than natural selection [17].

In this study, we combined genomic and evolutionary analyses to quantify how recent changes in lifestyle and culture might be affecting the underlying genetic basis of human sexually differentiated traits. First, we applied a genome wide association study (GWAS)-based approach to identify genetic variants associated with sexually differentiated phenotypes based on data from the UK Biobank study [18]. With a very large number of UK Biobank participants (~361,000 individuals in the dataset we analyzed), this analysis represents a powerful extension of several previous GWAS-based analyses of the genetic architecture of human sexually differentiated anthropometric traits [19]. We then used the Singleton Density Score (SDS), a statistic that identifies signatures of polygenic adaptation that acted within the last ~3,000 years [20], a period following the transition to agriculture in present-day United Kingdom [21].

## Results

We analyzed genome-wide association study (GWAS) summary statistics for the following five sexually differentiated anthropometric phenotypes that were produced by the Neale Lab [22] using data from the UK Biobank [18]: height, body mass, hip circumference, body fat percentage, and waist circumference. We chose these traits given their relevance to the motivating evolutionary hypothesis and because they have been extensively studied from anthropological and/or genomics perspectives [9,23–25]. We did not include body mass index (BMI) in our set of body size and shape phenotypes following concerns that have been voiced about this metric related to its failure to quantify body shape when indicating obesity and obesity-related health risks [26,27] and other, ethical issues [28]. We instead included the two constituents of BMI (height and body mass) as separate variables in our analysis.

### Identification of phenotype-associated SNPs

We analyzed summary statistics from GWAS that were performed separately in ~194,000 females and ~167,000 males of white British genetic ancestry on ~13.8 million autosomal SNPs [22]. SNPs with minor allele frequencies < 0.05 and low imputation quality were filtered out. We further restricted our analysis to SNPs that 1) passed these filters in both females and males and 2) had SDS values available [20], given our motivation to conduct downstream evolutionary analyses. These filtering steps resulted in ~4.4 million genome-wide SNPs for each sex-stratified GWAS. For each phenotype, we identified significantly phenotype-associated SNPs present in females, males, or both using the genome-wide significance threshold of $P = 5 \times 10^{-8}$ commonly applied in UK Biobank studies [29–31]. We identified the following total (not yet pruned for linkage disequilibrium) numbers of phenotype-associated SNPs significant in either females, males, or both: 67,738 for height, 15,669 for body mass, 12,580 for hip circumference, 10,538 for body fat percent, and 7,674 for waist circumference (Fig 1A and 1B and S1 Table).

### SNPs disproportionately associated with female or male trait variation

From the sets of phenotype-associated SNPs that were significant in females, males, or both for each phenotype (SexDiff-associated SNPs), we identified those SNPs with significant differences in the statistical strengths of association for the female vs. male-specific GWAS results using the t-SexDiff statistic [19]. This statistic estimates the probability of difference between female-specific and male-specific effect sizes given their standard errors and considering the genome-wide correlation between female and male effect sizes for each trait (see Methods). To account for multiple testing, we performed this analysis with four different false-discovery rate (FDR) cutoffs for each phenotype: 0.05, 0.01, 0.005, and 0.001.

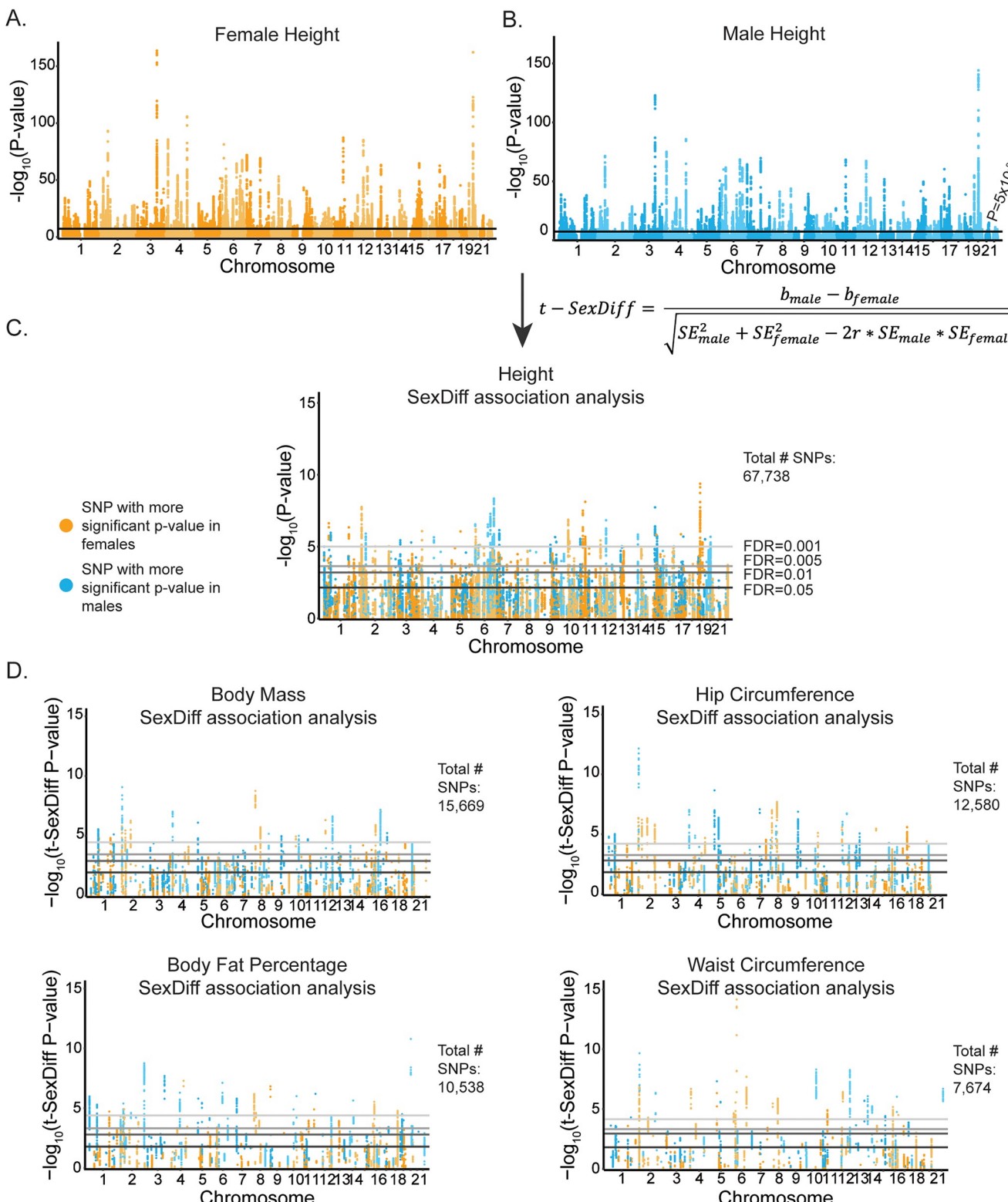

**Fig 1. SexDiff-associated SNPs for five anthropometric phenotypes.** (A) Manhattan plot depicting -log$_{10}$ P-values for the association of each genome-wide SNP with female height. The black line corresponds to the genome-wide significance threshold of P = 5x10$^{-8}$. (B) Manhattan plot for SNP associations with male height. (C) For each of the 67,738 SNPs significantly associated with female and/or male height, we used the equation shown to test whether the SNP was disproportionately associated with height between the sexes (height SexDiff-associated SNPs). The plot depicts -log$_{10}$ P-values for the t-SexDiff statistic. Gray

bars correspond to four different FDR cutoffs. (D) SexDiff association analyses for significant phenotype-associated SNPs (number of SNPs included in each analysis is shown to the right of each plot) for four additional anthropometric traits: body mass, hip circumference, body fat percentage, and waist circumference.

At the most stringent FDR cutoff of 0.001, we identified the following number of SexDiff-associated SNPs: 677 for height, 541 for body mass, 808 for hip circumference, 439 for body fat percentage, and 551 for waist circumference (Fig 1C and 1D and S1 Table). In addition to identifying SexDiff-associated SNPs, for each phenotype we calculated the proportion of Sex-Diff-associated SNPs at the FDR threshold of 0.001 to the number of phenotype-associated SNPs, which ranged from 0.0010 for height to 0.0718 for waist circumference.

## Co-localization of SexDiff-associated SNPs and loci with known roles in sexual differentiation

Prior to conducting evolutionary analyses on the anthropometric trait SexDiff-associated SNPs, we assessed their biological plausibility in two ways. First, we tested whether these SNPs are significantly more likely to be located within or nearby (+/- 10,000 base pairs) genes previously known to be involved in sexual differentiation (Gene Ontology term GO:0007548) compared to SNPs that are significantly associated with the phenotype but not significantly associated with sex differences [32,33]. Although not all SexDiff-associated SNPs are expected to be located nearby genes already known to be involved in sexual differentiation, we would expect at least some enrichment if we are identifying true SexDiff associations with our approach.

For each t-SexDiff FDR threshold, we determined the number of unique sexual differentiation (GO:0007548) genes with one or more co-localized SexDiff-associated SNPs (combined across the five phenotypes in our study). We similarly identified the number of all genes (of those in the GO database) that were co-localized with at least one SexDiff-associated SNP and calculated the proportion of the number of GO:0007548 genes to the total number of genes. We conducted the same analysis for the set of SNPs associated with the phenotypes in general but *not* associated with sexual differentiation. Finally, we estimated the GO:0007548 enrichment ratio for SexDiff-associated to non-SexDiff-associated SNPs by dividing the two proportions. By focusing this analysis on counts of genes rather than counts of SNPs, we limit potential linkage disequilibrium (LD)-based enrichment inflation. That is, a gene would only be counted once regardless of how many SexDiff-associated SNPs are located within or nearby that gene.

For example, at our most stringent FDR threshold (0.001), the 3,016 total SexDiff-associated SNPs were located within or nearby 9 unique GO:0007548 genes and 162 total genes (data included in Dryad Digital Repository deposition). Thus, the proportion of GO:0007548 genes = 0.0556 (9/162). In contrast, the 108,882 non-SexDiff-associated but still phenotype-associated SNPs for this same FDR threshold were located within or nearby 52 unique GO:0007548 genes and 2,544 total genes, for a proportion of 0.0204, resulting in an enrichment ratio = 2.73 (0.0556/0.0204). In other words, GO:0007548 genes with known roles in sexual differentiation are >2.7 times more likely to be co-localized with one or more SexDiff-associated SNPs than with one or more non-SexDiff-associated but still phenotype-associated SNPs at the FDR<0.001 analysis level.

We repeated this analysis for each FDR threshold (Fig 2A and S2 Table). The enrichment ratio steadily increased with increasingly stringent FDR significance thresholds. This pattern is consistent with expectations if our analyses are identifying true SexDiff-associated loci.

To test whether the proportion of GO:0007548 genes co-localized with at least one SexDiff-associated SNP was significantly greater than expected by chance, we used a simple permutation scheme. From the set of 2,570 total GO classified genes that were co-localized with at least one phenotype-associated SNP, we randomly selected the same number of total genes that were co-localized with one or more SexDiff-associated SNPs at the FDR cutoff being considered (e.g. 162 genes at FDR<0.001) and counted the number of GO:0007548 genes represented. We repeated this process 10,000 times for each FDR cutoff and compared the resulting distributions to the actual number of GO:0007548 genes co-localized with at least one SexDiff-associated SNP to calculate empirical P-values. We used this permutation-based approach rather than a Fisher's exact test because some genes were co-localized with both SexDiff-associated SNPs and non-SexDiff phenotype-associated SNPs. The observed proportion of GO:0007548 genes is very unlikely to be explained by chance at the most stringent FDR<0.001 threshold (P = 0.0041; Fig 2B), with increasing probability for the FDR thresholds of 0.005, 0.01, and 0.05 (P = 0.0513, P = 0.0829, and P = 0.2111, respectively; S1 Fig). Accordingly, all subsequent analyses were limited to SNPs classified based on the FDR<0.001 analysis.

In a second assessment of the biological plausibility of our results, we queried the GWAS Catalog [34] to identify any pleiotropic co-occurrence between our SexDiff-associated (FDR<0.001) or phenotype-associated SNPs and significant associations for all other GWAS Catalog traits. To avoid linkage disequilibrium (LD)-based result inflation of co-occurrence, we selected a maximum of one SNP (zero if no SNPs in the region were significantly associated with a trait) with the lowest t-SexDiff P-value across any of the five studied traits for each of 1,703 approximately LD-independent blocks of the human genome [35]. Of the 693 total SNPs in this curated dataset, 117 SNPs (17%) were significantly associated with sex differences for at least one of the five traits, with the remaining 576 SNPs (83%) associated with one or more of the five phenotypes but not sex differences.

Upon reviewing the overlap with the curated set of SexDiff-associated SNPs and the GWAS Catalog trait associations, we immediately noticed "Sex hormone binding globulin level" annotations [36–38] for 3 of the 117 SexDiff-associated SNPs (2.6%), including the two SNPs with the most extreme t-SexDiff P-values (Fig 2C). In contrast, sex hormone binding globulin level associations were annotated for 4 of the 576 phenotype- but not SexDiff-associated SNPs (0.7%; Fisher's exact test; P = 0.098). We also used a permutation analysis to estimate the probability of observing 3 SNPs with both SexDiff- and sex hormone binding globulin level-associations, given the occurrence rate for all loci in this curated SNP subset (P = 0.063; Fig 2D).

Altogether, these results suggest that the anthropometric trait SexDiff-associated SNPs we identified–especially those discovered with the most stringent FDR<0.001 threshold–are at least enriched for SNPs that do underlie (or are linked to those that do) sexual differentiation variation.

## SexDiff-associated SNPs are associated with greater trait effect sizes in the expected sex

Our statistical approach technically identifies genetic variants with significantly different strengths of association for a given trait between males and females. Before proceeding, we sought to confirm that these identified variants are disproportionately associated with greater phenotype effect sizes in the sex with the stronger association value, and not merely with lower trait value variance.

To do so, for each phenotype we first divided our SexDiff-associated SNPs into those with disproportionately stronger associations in females vs. males. To avoid linkage disequilibrium (LD)-based result inflation, we pruned each of these sets of SNPs to include only the one SNP

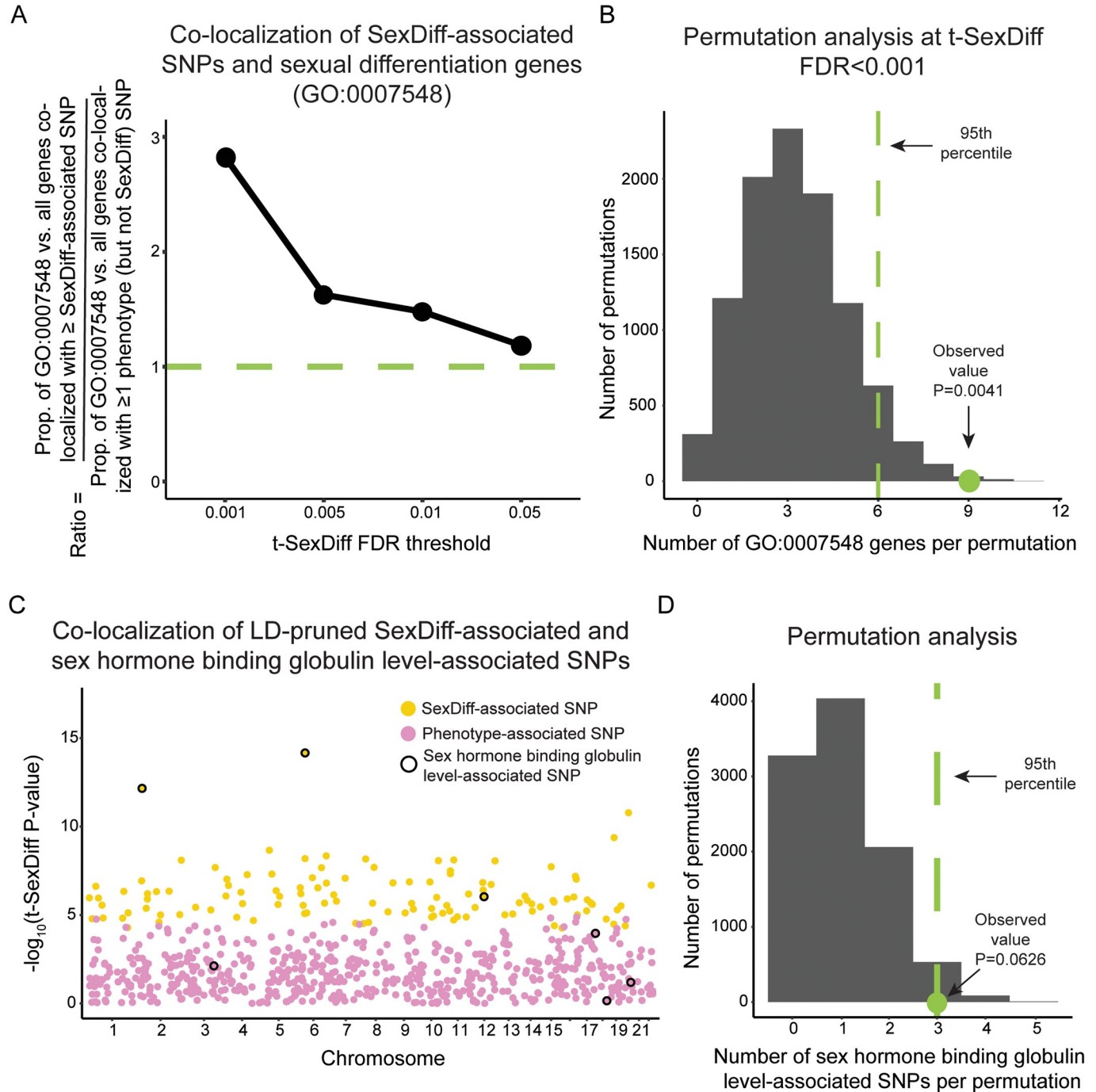

**Fig 2. Co-localization of SexDiff-associated SNPs and loci with known roles in sexual differentiation.** (A) For each t-SexDiff FDR threshold, we computed the proportion of the number of genes in the "Sexual Differentiation" Gene Ontology category (GO:0007548) to the number of all Gene Ontology genes with at least one co-localized SexDiff-associated SNP (+/- 10,000 base pairs). We also computed the same proportion for the set of SNPs significant associated with our studied phenotypes in general but *not* with sexual differentiation. Values shown are ratios of these two proportions at each t-SexDiff FDR threshold. The green line indicates the 1:1 ratio that would be expected in the absence of any disproportionate co-localization between SexDiff-associated SNPs and GO:00007548 genes. (B) Permutation analysis of the number of genes involved in sexual differentiation co-localized with at least one SexDiff-associated SNPs at our most stringent FDR threshold (q<0.001). From the set of 2,570 total GO-classified genes that were co-localized with at least one phenotype-associated SNP, we randomly selected 162 genes, the number of total genes that were co-localized with one or more SexDiff-associated SNPs at the FDR<0.001 cutoff. Of these 162 genes, we counted and recorded the number of GO:0007548 genes represented. We repeated this process 10,000 times and computed an empirical P-value (P = 0.0041) as the proportion of permutations with a greater than or equal number of GO:0007548 genes as the observed value for FDR<0.001 SexDiff-associated SNPs (9 genes). Results for similar analyses based on SexDiff FDR thresholds 0.005, 0.01, and 0.05 are shown in S1 Fig. (C) Manhattan plot depicting the -log10 values for the t-SexDiff statistic for a curated set of LD-pruned SexDiff and

phenotype-associated SNPs. Specifically, we selected a maximum of one SNP with the lowest t-SexDiff P-value across any of the five studied phenotypes from each of 1,703 approximately LD-independent blocks of the human genome. Yellow dots indicate SexDiff-associated SNPs at an FDR of 0.001 (for at least one of the five traits), and pink dots indicate phenotype-associated SNPs that did not cross the FDR = 0.001 threshold for any trait. SNPs outlined in black are also significantly associated with the sex hormone binding globulin levels as annotated in the GWAS catalog. (D) From the curated set of 693 total LD-pruned SexDiff and phenotype-associated SNPs, we randomly selected 117 SNPs (the number of pruned SexDiff-associated SNPs) and counted how many were also associated with sex hormone globulin blinding levels. We repeated this process 10,000 times and computed an empirical P-value of P = 0.0626.

(if any) with the lowest t-SexDiff FDR value located within each of 1,703 approximately LD-independent blocks of the human genome [35]. For comparison to the disproportionate female and male SexDiff-associated SNPs, we prepared similar LD-pruned sets of all SNPs significantly associated with the phenotype, regardless of their associations with sex differences, for each of the five traits (data included in Dryad Digital Repository deposition). Then, for every SNP in each set of pruned SNPs, we calculated the $\log_2$ ratio of the female trait effect size to the male trait effect size.

Each of the five sets of female SexDiff-associated SNPs had strongly positive average ratios, meaning that the female effect sizes for these SNPs were larger than the male effect sizes, and *vice versa* for the male SexDiff-associated SNPs (Fig 3A). Mean $\log_2$ ratios were significantly different from 0 for each of the 10 SexDiff-associated SNP subset (one-sided t-tests; $P = 3.3\times10^{-5}$ and lower; $FDR = 3.3\times10^{-5}$ and lower; S3 and S4 Tables). Conversely, the mean $\log_2$ ratios for the five sets of general trait-associated SNPs were near 0 (Fig 3A). Mean $\log_2$ ratios for each of the 10 SexDiff-associated SNPs were also significantly different from the corresponding mean $\log_2$ ratios of the phenotype-associated SNPs (Permutation analyses [see Methods]; P<0.001, FDR = 0.001). These results confirm that our statistical process successfully and reliably identifies SNPs with sex disproportionate effects on phenotypic trait variation. We thus proceeded to study the evolutionary histories of these SNPs.

## Signatures of positive selection on SexDiff-associated alleles

We next tested whether the SexDiff-associated SNPs identified in our analyses are significantly enriched for signatures of recent (~3,000 years) positive selection, using the Singleton Density Score statistic (SDS) [20]. Briefly, alleles affected by recent positive selection are predicted to be found on haplotypes with relatively fewer singleton mutations; the SDS quantifies this pattern. In turn, the trait-SDS statistic reflects directionality with respect to an associated phenotype. A positive trait-SDS value reflects a recent increase in the frequency of a phenotype-increasing allele, while a negative trait-SDS value reflects an increase in the frequency of a phenotype-decreasing allele [20]. We performed our analyses with previously-published SDS data computed from the whole genome sequences of 3,195 British individuals from the UK10K project [20,39]. This dataset is ideal for integration with the UK Biobank genotype-phenotype association data, given study population similarity.

To examine the evolutionary histories of alleles associated with anthropometric trait sex differences, we aimed to determine whether any signatures of positive selection for SexDiff-associated SNPs differed in either magnitude or direction from any such signatures for alleles associated with the phenotypes (but not necessarily sex differences) themselves. For this analysis, we again considered the sets of LD-pruned SexDiff-associated SNPs divided into those that are disproportionately associated with trait variation in females vs. those disproportionately associated with trait variation in males for each phenotype.

Using a permutation analysis, we tested whether the average trait-SDS values for the pruned female or male SexDiff-associated SNPs were significantly different than the value for the corresponding pruned set of all phenotype-associated SNPs (Fig 3B and S5 Table). For example, there were n = 21 pruned SexDiff-associated SNPs disproportionately associated with female height

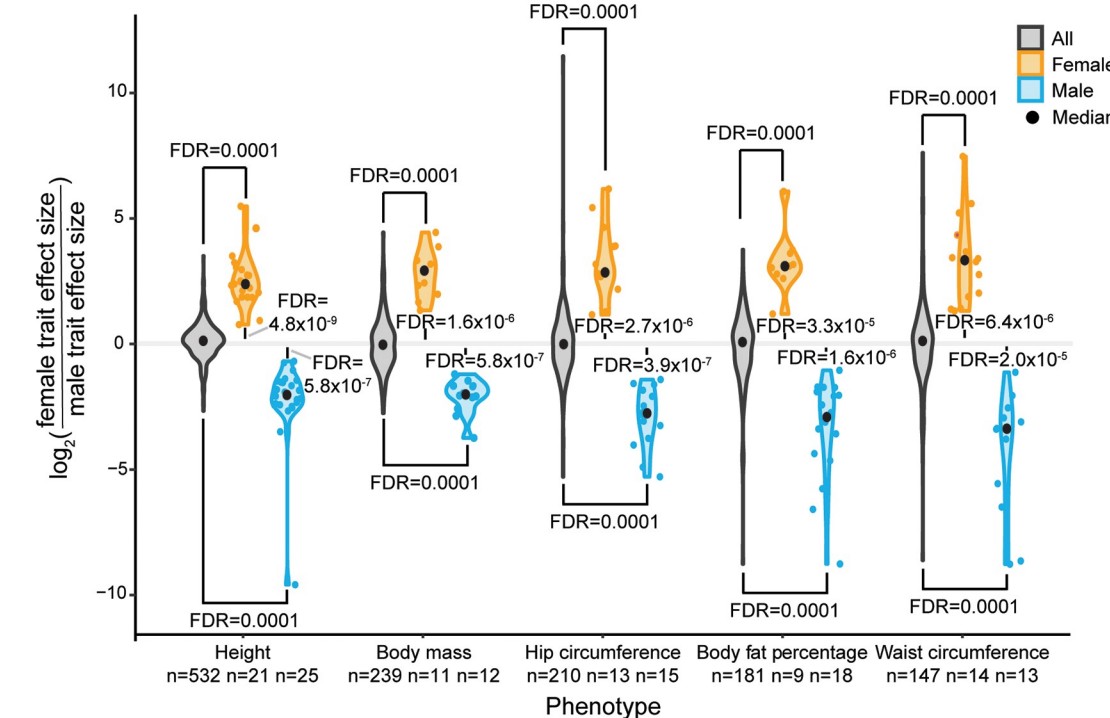

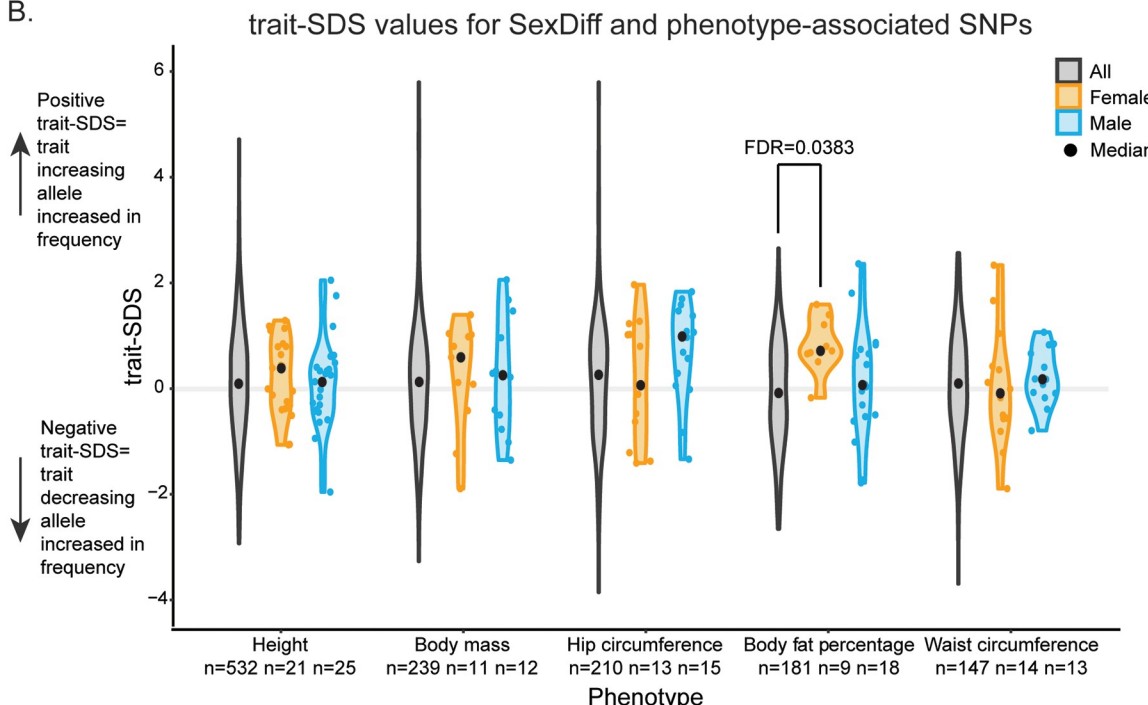

**Fig 3. Sex-specific effect size ratios and trait-SDS scores for anthropometric trait-associated SNPs.** For each anthropometric trait, a maximum of one SNP per each of 1,703 approximately LD-independent blocks of the human genome was included (within each block, the SNP with the strongest statistical significance) from the sets of i) all SNPs associated with trait variation, and the subsets of those SNPs disproportionately associated with variation in ii) females and iii) males (SexDiff-associated SNPs at FDR<0.001). (A) Violin plots of $\log_2$ ratios of female trait effect size to male effect size (calculated separately for each SNP). Mean $\log_2$ ratios for each set of female and male SexDiff pruned SNPs were compared to those for the corresponding phenotype-association set using a permutation scheme (FDR

values indicated at the top and bottom of the plot), and to 0 using a one-sided t-test (FDR values indicated in the middle of the plot). (B) Violin plots of trait-SDS values. Positive trait-SDS values reflects recent increases in the frequencies of phenotype-increasing alleles, while negative trait-SDS values reflect increases in the frequencies of phenotype-decreasing alleles. The trait-SDS distributions for each set of female and male SexDiff pruned SNPs were compared to those for the corresponding phenotype-association set using a permutation analysis.

and n = 532 pruned SNPs associated with height generally (but not necessarily with sex differences). We randomly selected 21 of the 532 pruned height-associated SNPs and calculated the average trait-SDS score. We repeated this process 10,000 times and calculated an empirical P-value based on the proportion (multiplied by two; see Methods) of permuted observations with equal or more extreme average trait-SDS values than the observed average trait-SDS for the actual female height SexDiff-associated SNPs (average trait-SDS = 0.240; P = 0.66). This test was then repeated for the male SexDiff-associated SNPs for height, and the female and male SexDiff-associated SNPs for each of the remaining four anthropometric traits. We also performed a similar permutation but with the additional step of matching minor allele frequencies (+/- 0.05) between the original set of SexDiff-associated SNPs and the set of phenotype-associated SNPs selected in each permutation. The pattern of results was unchanged relative to the above (S6 Table).

For nine of the comparisons, trait-SDS distributions for the sets of pruned SexDiff-associated SNPs were not significantly different (following FDR adjustment for ten tests) from those for SNPs associated with the corresponding general phenotype (Fig 3B and S5 Table). However, compared to all body fat percentage-associated SNPs (n = 181 pruned SNPs; average trait-SDS = -0.124), the average trait-SDS value for the set of female SexDiff-associated SNPs for this phenotype was significantly elevated (n = 9 pruned SNPs; average trait-SDS = 0.827; P = 0.0038; FDR = 0.038). In other words, the average frequencies of alleles associated with greater body fat percentage in females have increased in frequency over the past ~3,000 years at a faster rate than expected based on the pattern for SNPs associated with body fat percentage generally, a potential signature of polygenic selection on this sub-trait.

While our hypothesis testing framework is focused primarily on analyses of recent signatures of positive natural selection, we also investigated whether the sets of LD-pruned SexDiff-associated SNPs were enriched for signatures of relatively more ancient positive selection. To do so we conducted an identical permutation analysis as above with the trait-SDS values, but instead using absolute values of the integrated haplotype score (iHS) statistic, a within-population haplotype-based statistic that can be used to identify putative signatures of positive selection from across the past ~25,000 years [40]. We used previously-published iHS data [41] computed for the Great Britain population (GBR) from the 1000 Genomes Project [42]. Absolute iHS values were not significantly different for any of the sets of SexDiff-associated SNPs relative to the corresponding phenotype-associated SNPs (S2 Fig and S7 Table).

Finally, because positive selection may occur more often within or nearby genic regions of the genome [40], we also tested whether our sets of pruned SexDiff-associated SNPs and corresponding phenotype-associated SNPs were unevenly distributed among genic (annotated genes +/- 10,000 bp) and intergenic regions. None of the ten sets of SexDiff-associated SNPs were enriched for genic regions. However, we found that SNPs significantly associated with male waist circumference variation were more likely to be intergenic relative to general hip circumference-associated loci (S8 Table).

## Discussion

Using a sex-stratified GWAS framework for five sexually differentiated anthropometric phenotypes, we identified 3,016 SNPs that were disproportionately associated with either female

or male trait variation at a low false discovery rate (FDR<0.001). We confirmed the biological plausibility of these results by showing that genes with known roles in sexual differentiation are significantly enriched for SexDiff-associated SNPs. Together, these results confirm the importance of considering sex differences when investigating the genetic structure of human polygenic traits [43]. We then used a statistic that quantifies changes in the frequencies of alleles underlying polygenic traits over the past ~3,000 years to identify a signature of recent positive selection on SNPs associated with increased female body fat percentage in the British study population.

We must emphasize that inferring selection signals from GWAS data should be approached with great care, as even subtle uncorrected population structure can impact GWAS and down-stream results [44]. For example, data from the GIANT consortium were previously used to identify strong signatures of polygenic selection for height across the genome [20]. However, subtle population structure in the GIANT sample led to effect-size estimate biases, in turn resulting in false signals of polygenic selection for SNPs not crossing the genome-wide significance threshold and impacting results for significant SNPs as well [44]. In contrast, these issues were much less prevalent using GWAS summary statistics from the UK Biobank, in which population structure is minimized [44–46]. In light of these considerations, in our study we have i) used UK Biobank GWAS summary statistics only, ii) focused solely on phenotype-associated SNPs below the genome-wide significance threshold, and iii) restricted our evolutionary analyses to direct comparisons between SNPs significantly associated with individual phenotypes and a sub-phenotype (i.e., sex differences).

Our study further demonstrates the value of GWAS-based approaches for testing anthropological hypotheses [47]. Concerning the evolution of human body size and shape phenotypes, our results fail to provide support for the prevailing notion of recent (i.e., subsequent to agriculture) adaptive reductions in levels of sex differences for such traits. Specifically, using large samples of genomes from British individuals we did not observe significant differences in the recent evolutionary trajectories of SNPs disproportionately associated with female or male variation in height, body mass, hip circumference, and waist circumference relative to the trajectories of SNPs associated with these traits generally.

We note that we made a number of conservative choices (for example, with aggressive pruning to account for linkage disequilibrium) in our analytical approach, meaning that our failure to reject the null hypothesis for each of these four traits should not be interpreted as evidence that no selection on them occurred. Still, even with our conservative analytical approach we did find evidence that the average frequencies of alleles disproportionately associated with greater female body fat percentage significantly increased over the past ~3,000 years, a pattern consistent with polygenic adaptation. Given that females have higher average body fat percentages than men in historic and contemporary populations, the direction of polygenic adaptation in the population we studied would actually be opposite to expectations under hypotheses of recent adaptive reductions in anthropometric trait sex differences in agricultural societies. However, since SNPs can be pleiotropically associated with multiple phenotypes [35], we cannot definitively conclude that positive selection acted directly on female body fat percentage. Regardless, at the very least we did not find positive support for the prevailing hypothesis concerning the evolution of sex differences in recent human evolution.

## Methods

### Subjects and dataset generation

We used genome-wide association study (GWAS) summary statistics generated from analyses of UK Biobank data [18,22]. The original GWAS analyses were restricted to 361,194 unrelated

individuals (194,174 females and 167,020 males) of white British ancestry (based on the combination of self-report and genetic ancestry analysis) who did not have sex chromosome aneuploidies. GWAS summary statistics for each phenotype were computed separately for females and males. Because every phenotype was not defined for all individuals, some of the analyses contributing to our study included fewer than 361,194 individuals (S9 Table). Approximately 40 million SNPs were originally available for analysis, of which only those with minor allele frequencies > 0.001, an INFO score (imputation quality) > 0.8, and a Hardy-Weinberg Equilibrium P-value > $1 \times 10^{-10}$ were retained, resulting in datasets of 13,791,468 SNPs.

We then subsequently filtered out SNPs with minor allele frequencies < 0.05 and those that were not associated with a singleton density score (SDS; for description, see below) [20]. These filters resulted in a total genome-wide dataset of ~4.4 million SNPs for each phenotype.

We chose five sexually differentiated anthropometric phenotypes from the GWAS summary statistics for analysis (S9 Table). To identify phenotype-associated SNPs in female and/or male individuals, we applied the commonly used genome wide significance threshold (i.e., to account for the large number of tested SNPs) of P = $5 \times 10^{-8}$ to the male-specific and female-specific GWAS summary statistics (data included in Dryad Digital Repository deposition).

## Scan for SNPs that are disproportionately associated with male or female trait variation

For each genome-wide SNP significantly associated with a phenotype in females, males, or both sexes, we evaluated whether there was a significant difference in the statistical strengths of association for the female vs. male-specific GWAS results using the following t-statistic (t-SexDiff) [19].

$$t - SexDiff = \frac{b_{male} - b_{female}}{\sqrt{SE_{male}^2 + SE_{female}^2 - 2r * SE_{male} * SE_{female}}}$$

$b_{male}$ refers to the male-specific beta value and $b_{female}$ refers to the female-specific beta value for each genome-wide SNP. $SE_{male}$ and $SE_{female}$ refer to the male-specific and female-specific standard error for each genome-wide SNP, respectively. The correlation *r* (rho) for each phenotype was calculated as the Spearman rank correlation coefficient between $b_{male}$ and $b_{female}$ using the cor.test() function in R (version 3.5.1). t-SexDiff was converted to a two-sided P-value using the R function pt(). The effects of multiple testing were considered by computing the False Discovery Rate (FDR) for each t-SexDiff P-value using the R function p.adjust [48].

Of the phenotype-associated SNPs that were significantly associated with a given phenotype for females, males, or both sexes, we identified SexDiff-associated SNPs at four different FDR thresholds: <0.05, <0.01, <0.005, and <0.001 (data included in Dryad Digital Repository deposition).

## Assessing the biological plausibility of the SexDiff-associated SNPs

In order to confirm the biological plausibility of our SexDiff-associated SNPs, we tested whether regions of the genome functionally linked to sexual differentiation are more likely than expected by chance to contain one or more of our SexDiff-associated SNPs. Separately for each t-SexDiff FDR threshold, we counted the number of unique genes in the Gene Ontology database [32,33] that contained at least one SexDiff-associated SNP, including within a +/-10,000 base-pair (bp) window around the gene to encompass potential regulatory regions. We then counted the number of these genes with known links to processes of sexual differentiation corresponding to the Gene Ontology (GO) term GO:0007548 and computed the

proportion of the number of GO:0007548 genes to the number of all genes co-localized with ≥ 1 SexDiff-associated SNP. We repeated this analysis for significant phenotype- but not Sex-Diff-associated SNPs. In the absence of enrichment for SexDiff-associated SNPs, the ratio of these two proportions is expected to equal one.

We then used the following permutation scheme for each t-SexDiff FDR cutoff. There was a total of 2,570 GO-classified genes overlapping one or more phenotype-associated SNP (whether SexDiff-associated or not; each t-SexDiff FDR cutoff started with the same genome-wide significant set of phenotype-associated SNPs so the total number of 2,570 co-localized genes applies to each FDR cutoff). Given the number of these genes overlapping ≥ 1 SexDiff-associated SNP for a given FDR cutoff, we randomly selected that number of unique genes from the pool of 2,570 genes and counted the number of GO:0007548 genes. We then repeated this procedure 10,000 times and computed an empirical P-value as the proportion of permuted data sets with an equal or greater to number of sexual differentiation genes when compared to our observation for the associated FDR threshold.

We studied pleiotropic relationships between our anthropometric SexDiff and phenotype traits and other traits with the aid of the GWAS Catalog (48; accessed 18 Feb 2021). After concatenating all SexDiff (FDR<0.001) and phenotype-associated SNPs from across the five studied traits, we pruned this total set of SNPs to a maximum of one SNP per each of approximately 1,703 approximately LD-independent blocks of the human genome [35]. For each block, we specifically chose the one SNP with the lowest t-SexDiff P-value (if a block did not have any significant phenotype-associated SNPs for any trait, zero SNPs were included from that block). There were 693 total SNPs in this curated dataset, 117 of which (17%) were significantly associated with sex differences for at least one of the 5 traits, with 576 (83%) significantly associated with one or more of the five phenotypes but not sex differences. After reviewing the full results (data included in Dryad Digital Repository deposition), we used a permutation analysis to estimate the probability that SexDiff-associated SNPs were more likely than expected by chance to also be associated with the "sex hormone binding globulin level" trait. Specifically, from our curated set of 693 SNPs, we randomly selected the number of SNPs equal to the number of observed SexDiff-associated SNPs (117) and counted how many were also associated with sex hormone binding globulin level as per the GWAS catalog. We repeated this procedure 10,000 times and computed the proportion of permuted data sets with equal or more extreme numbers of sex hormone binding globulin level-associated SNPs compared to the result for the SexDiff-associated SNPs.

## Assessing the sex-specific effects of SexDiff-associated SNPs

For each phenotype we split our SexDiff-associated SNPs into those that had lower P-values (as identified in the original sex-specific GWAS for phenotype-associated SNPs) in females than males (female SexDiff-associated SNPs) and those that had lower P-values in males than females (male SexDiff-associated SNPs). We separately pruned each set of female SexDiff-associated SNPs and male SexDiff-associated SNPs to account for linkage disequilibrium. Specifically, if there was more than one female SexDiff-associated SNP in one of 1,703 approximately LD-independent segments of the genome [35], we only kept the female SexDiff-associated SNP for that segment with the lowest P-value for association with the phenotype in females. We did the same for male SexDiff-associated SNPs in males. For each set of pruned SexDiff-associated SNPs, we then calculated the $\log_2$ ratio of the female effect size to male effect size.

The set of phenotype-associated SNPs was pruned in a similar fashion to the above, with a maximum of one SNP per each of the 1,703 approximately LD-independent genome segments, chosen as the SNP with the most significant P-value in the segment regardless of whether it

was most significant in females or males (data included in Dryad Digital Repository deposition) from among those below the genome-wide significance threshold.

We then used a permutation to estimate the probability that the $\log_2$ ratio of female effect size to male effect size was significantly for each of the ten sets of SexDiff-associated SNPs. From the pruned set of phenotype-associated SNP, we randomly selected a number of SNPs equal to the number of observed SexDiff-associated SNPs for that phenotype. We repeated this procedure 10,000 times and computed the proportion of permuted data sets with equal or more extreme average $\log_2$ ratio. We computed FDR values to account for multiple testing.

Finally, we used a one-sided t-test to determine whether our $\log_2$ ratios for each set of SexDiff-associated SNPs was significantly different from zero. We again computed FDR values to account for multiple testing.

## Identification of signatures of positive selection

To test the hypothesis that SexDiff-associated SNPs have been affected by recent (past ~3,000 years) positive selection in recent human evolution, we used the Singleton Density Score (SDS) statistic [20]. We used a database of genome-wide SNP SDS scores that were computed [20] using 3,195 whole genome sequences from British individuals the UK10K project [39]. For the pruned sets of female and male SexDiff-associated SNPs, we fixed the sign of SDS scores so that positive values indicate an increased frequency of the trait-increasing allele.

We used permutations to estimate the probability that the average trait-SDS value for each trait and sex could be observed by chance given the distribution trait-SDS scores for SNPs significantly associated with the corresponding phenotype (regardless of SexDiff-association; S10 Table). From the pruned set of phenotype-associated SNPs, we then randomly selected a number of SNPs equal to the number of observed SexDiff-associated SNPs for that phenotype and the sex and FDR threshold being considered, and we calculated the average trait-SDS score for that set of SNPs. We repeated this procedure 10,000 times and computed the proportion of permuted data sets with equal or more extreme average trait-SDS scores compared to actual result for the observed SexDiff-associated SNPs. This proportion was then multiplied by two to account for the two-tailed nature of this test (i.e., the average trait-SDS values for SexDiff-associated SNPs could have been significantly greater than or less than that for the phenotype-associated SNPs). We computed FDR values to account for multiple testing.

We also performed trait-SDS permutation analysis with minor allele frequency (MAF) matching between SexDiff and phenotype-associated SNPs. Specifically, for each SexDiff-associated SNP we identified all phenotype-associated SNPs with +/- 0.05 MAF. We then randomly selected one of these SNPs for inclusion in the permuted dataset and removed it from our list of phenotype-associated SNPs to be drawn from in the MAF matching to other SexDiff-associated SNPs so that it wouldn't be represented twice in the same permutation. We repeated this procedure 10,000 times, randomizing the input order of SexDiff-associated SNPs each time.

For our iHS-based analysis, we used a database of genome-wide SNP iHS scores that were previously computed [41] using data from the 1000 Genomes project [42]. We specifically used iHS data from the Great Britain (GBR) population due to population similarity with our GWAS data. We considered the absolute value of the standardized iHS for each SNP, and we removed SNPs from our dataset that did not have a corresponding iHS score. To estimate the probability that the average iHS score for each trait and sex could be observed by chance given the distribution of iHS scores for phenotype-associated SNPs, we used a permutation scheme identical to that used for the primary trait-SDS analysis (the version without consideration of minor allele frequencies).

To test whether there were significant differences in the distributions of pruned SexDiff-associated SNPs and corresponding pruned phenotype-associated SNPs, we determined whether or not each SNP was located within +/- 10,000 bp of a gene annotated in the Gene Ontology database. We compared the ratio of genic:intergenic SNPs for each of our ten sets of SexDiff-associated and corresponding phenotype-associated SNPs for each trait using a permutation analysis (S8 Table). For each set of SexDiff-associated SNPs, we randomly selected the same number of SNPs from the corresponding set of phenotype-associated SNPs and counted the number of SNPs in intergenic regions. We then repeated this procedure 10,000 times and computed an empirical P-value as the proportion of permuted data sets with an equal or greater to number of SNPs in intergenic regions when compared to our observation for the associated set of SexDiff-associated SNPs. This proportion was then multiplied by two to account for the two-tailed test (i.e., the number of intergenic SexDiff-associated SNPs could have been significantly greater than or less than that of phenotype-associated SNPs). We computed FDR values to account for multiple testing.

## Computational resources

All analyses were conducted in R (version 3.5.1) with Advanced CyberInfrastructure computational resources provided by The Institute for CyberScience at Pennsylvania State University. All scripts are available at https://github.com/audreyarner/dimorphism-evolution.

## Supporting information

**S1 Fig. Permutation enrichment distribution at each FDR threshold.** Permutation analysis of the number of genes involved in sexual differentiation for all anthropometric SNPs at every FDR threshold. Data are the frequency of distribution of our results for 10,000 permuted data sets. The empirical P-value represents the probability that the observed value of sexual differentiation genes from our SexDiff-associated SNP pool is equal to or greater than those from a randomly selected set.
(TIF)

**S2 Fig. Sex-specific iHS scores for anthropometric SexDiff and phenotype-associated SNPs.** The iHS distributions for each set of female and male SexDiff pruned SNPs were compared to those for the corresponding phenotype-association set using a permutation analysis. None of the distributions were significantly different.
(TIF)

**S1 Table. Observed number of SexDiff-associated SNPs at each FDR threshold for every phenotype.** [a]Ratio of SexDiff-associated SNPs at the FDR threshold of 0.001 to the number of phenotype-associated SNPs.
(DOCX)

**S2 Table. Observed unique sexual differentiation genes (SDG) and total number of genes for SexDiff-associated SNPs and Non SexDiff-associated SNPs.** [a]Proportion of SDG genes observed in each SNP group [b]Ratio of proportion of SDG genes observed in the group of SexDiff-associated SNPs to the proportion of SDG genes observed in the group of non SexDiff-associated SNPs.
(DOCX)

**S3 Table. Observed log$_2$ ratio of female to male beta values and p-values for each set of Female SexDiff-associated SNPs.** [a]Number of pruned SexDiff-associated SNPs at an FDR threshold of 0.001 [b]Mean log$_2$ ratio of female trait effect size to the male trait effect size [c]One-

sided t-test P-value comparing distribution of the log$_2$(ratio) [d]Permutation P-value of the probability that the mean log$_2$(ratio) could be observed by chance when compared to phenotype-associated SNPs.
(DOCX)

**S4 Table. Observed log$_2$ ratio of female to male beta values and p-values for each set of Male SexDiff-associated SNPs.** [a]Number of pruned SexDiff-associated SNPs at an FDR threshold of 0.001 [b]Mean log$_2$ ratio of female trait effect size to the male trait effect size [c]One-sided t-test P-value comparing distribution of the log$_2$(ratio) [d]Permutation P-value of the probability that the mean log$_2$(ratio) could be observed by chance when compared to phenotype-associated SNPs.
(DOCX)

**S5 Table. Observed trait-SDS and permutation P-values for each set of Female SexDiff-associated SNPs and Male SexDiff-associated SNPs permuted against phenotype-associated SNPs.** [a]Number of pruned SexDiff-associated SNPs at an FDR threshold of 0.001 [b]Average trait-SDS score of pruned set of SexDiff-associated SNPs [c]Permutation P-value of the probability that the trait-SDS score for each sex could be observed by chance when compared to phenotype-associated SNPs.
(DOCX)

**S6 Table. Observed trait-SDS and permutation P-values for each set of Female SexDiff-associated SNPs and Male SexDiff-associated SNPs permuted against phenotype-associated SNPs matched for minor allele frequency.** [a]Number of pruned SexDiff-associated SNPs at an FDR threshold of 0.001 [b]Average trait-SDS score of pruned set of SexDiff-associated SNPs [c]Permutation P-value of the probability that the trait-SDS score for each sex could be observed by chance when compared to phenotype-associated SNPs matched for minor allele frequency.
(DOCX)

**S7 Table. Observed average |iHS| scores and permutation P-values for each set of Female SexDiff-associated SNPs and Male SexDiff-associated SNPs permuted against phenotype-associated SNPs.** [a]Number of pruned SexDiff-associated SNPs at an FDR threshold of 0.001 [b]Average |iHS| score of pruned set of SexDiff-associated SNPs [c]Permutation P-value of the probability that the |iHS| score for each sex could be observed by chance when compared to phenotype-associated SNPs.
(DOCX)

**S8 Table. Observed number of intergenic SNPs and permutation P-values for each set of Female SexDiff-associated SNPs and Male SexDiff-associated SNPs permuted against phenotype-associated SNPs.** [a]Number of pruned SexDiff-associated SNPs at an FDR threshold of 0.001 [b]Permutation P-value of the probability that the number of intergenic SNPs in each set of SexDiff-associated SNPs could be observed by chance when compared to phenotype-associated SNPs.
(DOCX)

**S9 Table. Phenotype information.** [a]As referred to in the Neale lab manifest released on July 31, 2018 [b] Correlation for each phenotype calculated as the Spearman rank correlation coefficient between beta values of men and women.
(DOCX)

**S10 Table. Observed trait-SDS for each set of pruned phenotype-associated SNP groups.** (DOCX)

## Acknowledgments

We thank our colleagues at the DFG Center for Advanced Studies "Words, Bones, Genes, Tools" for their support and S. Marciniak, C. Bergey, J. Tung, and D. Puts and the Sex Differences Interest Group for their helpful discussions.

## Author Contributions

**Conceptualization:** Audrey M. Arner, Kathleen E. Grogan, Mark Grabowski, Hugo Reyes-Centeno, George H. Perry.

**Data curation:** Audrey M. Arner.

**Formal analysis:** Audrey M. Arner.

**Funding acquisition:** Audrey M. Arner, Kathleen E. Grogan, Mark Grabowski, Hugo Reyes-Centeno, George H. Perry.

**Investigation:** Audrey M. Arner.

**Methodology:** Audrey M. Arner, Kathleen E. Grogan, Mark Grabowski, Hugo Reyes-Centeno, George H. Perry.

**Project administration:** Audrey M. Arner, Kathleen E. Grogan, George H. Perry.

**Resources:** Mark Grabowski, Hugo Reyes-Centeno, George H. Perry.

**Software:** Audrey M. Arner.

**Supervision:** Kathleen E. Grogan, George H. Perry.

**Validation:** Audrey M. Arner.

**Visualization:** Audrey M. Arner.

**Writing – original draft:** Audrey M. Arner, Kathleen E. Grogan, George H. Perry.

**Writing – review & editing:** Audrey M. Arner, Kathleen E. Grogan, Mark Grabowski, Hugo Reyes-Centeno, George H. Perry.

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
