## [Decision Letter · Decision Letter 0]

29 Dec 2020

Dear Dr Arner,

Thank you very much for submitting your Research Article entitled 'Patterns of recent natural selection on genetic loci associated with sexually differentiated human body size and shape phenotypes' to PLOS Genetics.

The manuscript was fully evaluated at the editorial level and by independent peer reviewers. The reviewers appreciated the attention to an important topic but identified some concerns that we ask you address in a revised manuscript

We therefore ask you to modify the manuscript according to the review recommendations. Your revisions should address the specific points made by each reviewer.

[LINK]

Yours sincerely,

Scott M. Williams

Section Editor: Natural Variation

PLOS Genetics

Hua Tang

Section Editor: Natural Variation

PLOS Genetics

Reviewer's Responses to Questions

**Comments to the Authors:**

Reviewer #1: The manuscript by Arner et al. tests whether genetic variants are associated with sexually differentiated phenotypes in humans using UKBioBank data. The authors first find SNPs associated with different traits in the UK Biobank. They then test whether the strength of the association differs between males and females. Indeed, the authors find that these SexDiff SNPs are enriched in genes involved in sexual differentiation. Lastly, the authors test whether the SexDiff SNPs show greater signals of recent positive selection, as compared to SNPs associated with the overall phenotype. The authors find that alleles associated with increased body fat percentage in only females tend to show signals of positive selection via the SDS statistic more so than SNPs associated with body fat percentage. The other traits examined do not show signals of such sexually differentiated positive selection.

Overall this paper addresses an interesting question that has not received as much attention in the empirical human genetics literature. The statistical analyses seem solid and carefully thought-out. I have several comments to help improve the manuscript:

Major Comments:

1) Starting on line 154 forward: In this section, the authors test whether the SexDiff SNPs are enriched in genes that have a known role in sexual differentiation. This is a nice addition to the paper and helps strengthen the case that these SNPs are biologically functional. Were the associated SNPs pruned for linkage disequilibrium for these analyses? It was unclear to me at which stage of the analyses the LD pruning was done. Unless I’m missing something (please correct me if I am), the LD pruning should be done prior to testing for enrichments as the enrichment tests are assuming that the SNPs are more or less independent. Further, as genes in different functional categories may have different recombination rates, patterns of LD, etc, it makes sense to include putatively independent SNPs as much as possible.

2) Figure 2 and associated analyses: I appreciate the resampling approach to test for enrichment. Was there a specific reason why the permutation analysis was used instead of Fisher’s exact test? It might be good to clarify more on this.

3) Section starting on line 222: I had a hard time understanding the statistical or biological relevance of this section. What is the purpose of testing for whether the SexDiff variants are disproportionately associated with greater phenotype effect sizes in the sex with the stronger association values? Does this not occur by design? Better motivating the purpose of this analysis would strengthen the manuscript in my view.

4) Figure 3 and tSDS analyses: For each trait, the authors compare the tSDS statistic distribution of the sexDiff SNPs to the tSDS statistic distribution for SNPs associated with the trait. Is this the right null distribution? What if the traits themselves are undergoing positive selection, just not in a sexually differentiated way? Then the SexDiff SNPs may not be expected to look different even if they are under selection too. Additionally, it’s not clear to me what the ideal null distribution for comparison would be in these cases. More explanation as to the rationale for the choice of the null set of SNPs would be good.

5) Figure 3 and tSDS analyses: Are the control SNPs for the tSDS statistic matched for allele frequency and recombination rate to the SexDiff SNPs in each comparison? It would be important to match for these putative confounders that differ across the genome and may influence the tSDS values.

6) Discussion: The authors included a nice discussion about population stratification as a potential confounder. I think it would be good to also discuss whether associative mating might be expected to influence any of these results. Further, on the other side, it would also be good to include some sort of discussion of the power of these analyses. It appears the authors made a number of conservative decisions in their statistical analyses (which I think is good, unless otherwise noted here) and it would be good to put this into context. For example, how strongly do the negative results reject the hypotheses of recent reductions in anthropometric trait sex differences in agricultural societies?

Minor comments:

1. Lines 36-38: This wording was a little confusing. Maybe say something like “SNPs associated with significant differences in association between males and females are enriched genes with known roles in sexual differentiation, as compared to other SNPs associated with the trait”.

2. Line 63: This seems like an odd start to the paper. Why not just start talking about “sexually differentiated traits” and their biology, rather than say that these are not “sexually dimorphic”. That statement can come elsewhere later in the Intro.

3. Lines 137: It would be good to give a little more context and description of the t-SexDiff test here. I realize that it’s fully explained in the Methods, but given that Results come first, I think it would be nice to give a short explanation to those who do not read the Methods.

4). Lines 142-144: Is this a ratio or a proportion?

5) Enrichment analysis in Figure 2: Were 162 or 135 genes selected? Both seem to be used (lines 191 and 206).

6) Figure 2B: To avoid confusion, the “differentiation” label on the x-axis might better be called “Number of GO:00007548 genes per permutation” or something like that. I worry that some may think “differentiation” might refer to the Fst or the population genetic version of differentiation, which of course is not what’s meant.

7) Line 293: Specify how many tests were corrected for with the FDR adjustment.

Reviewer #2: In this manuscript the authors are interested in two related questions: whether there is evidence of significant differential genetic architecture between human males and females across multiple complex traits and, if so, whether these differences can be explained by recent positive selection in the directions expected based on the first set of results. The authors are interested in this question in part due to anthropological theory that states the change in human history to subsistence farming from hunter and gatherers should be accompanied by a reduction in differentiating characteristics between males and females, as the necessary traits in males and females become more homogenous.

The authors approach these questions by analyzing white British individuals in the UK BioBank dataset across multiple complex traits. They conduct sex-differentiated genome-wide association studies and compare both the number of associations between the sexes and the strength of association. They also look at whether genes related to sexual differentiation are enriched among their top hits. Lastly, using their sex-differentiated association SNPs (sex-diff SNPs), they conduct positive selection analyses using the singleton density score, a method that can identify recent positive selection. The authors find evidence for both differential genetic architecture between males and females across the traits analyzed, as well as evidence for at least one trait and sex combination (body fat percentage in females) to be undergoing recent positive selection, though in the opposite direction (increasing in allele frequency) than one would expect from the anthropological theory.

Overall I think this is a good manuscript and study design. I think the analyses and justifications make sense, and the flow of the different parts work well. I also appreciate how the work shown here has a connection to a theory from anthropology, grounding the results as well as potentially broadening their impact too. I do have a few suggestions regarding some additional analyses that might further bolster the positive selection sections, as well as a few minor comments.

Major Comments:

1) For the SDS results, I am wondering whether the presence of older positive selection might affect the results somehow, or at least change our interpretation of the results. For instance, if we find that there is evidence for older positive selection on body fat percentage in females, then maybe our interpretation of the SDS results changes from recent selection due to the change in subsistence strategy to simply ongoing selection for sexual differentiation. In fact maybe we would expect some amount of sexual differentiation to be continuously selected for due to phenomena like assortative mating or sexual conflict. To get at this, could the authors run some other tests for positive selection, such as Berg and Coop’s Qs or statistics for selective sweeps? If there is evidence for older positive selection on the traits that did not show significant SDS results as well, I would wonder then if it’s possible that SDS might be less effective due to presumably lower levels of local genetic variation.

2) Similarly for the SDS results, is there any difference between the distribution of sex-diff SNPs and the non-sex-diff SNPs in terms of genic vs. intergenic regions? Once again I am wondering about other factors that may be affecting the results, and whether if lower levels of local genetic variation for the sex-diff SNPs might somehow impact the SDS results, ie if sex-diff SNPs were somehow more often in genic regions than non-sex-diff SNPs are. If this were the case then maybe some of the negative SDS results for the other traits are a result of being under powered.

3) The authors mention the possibility that pleiotropy could be occurring among the sex-diff SNPs showing evidence for positive selection in female body fat percentage. Since the authors have already used the summary statistics from the Ben Neale UKB GWAS results, I would be curious whether as a group these SNPs are also all associated with any other traits. If as a group we find a pattern of traits these SNPs are associated with, or on the other hand find no evidence these SNPs are associated with any other traits (at least among the traits previously analyzed), that could be an interesting additional piece of information to help with the interpretation.

Minor Comments:

1) For Figure 1 I have a few comments, mostly related to consistency. I don’t believe the red vs. pink motif in 1A continues throughout the rest of the figure (likely due to the combination of red and yellow later on), so I’m not sure if it should be used at all. The Bonferroni p-value threshold line should be labeled in 1A and 1B since the FDR lines are labeled later on as well. And in the legend the FDR lines are described as ‘green bars’, though in the copy of the manuscript I have, the bars are blue. This might just be my copy, but it should be checked and corrected if needed.

2) For lines 233-235, I think “...we prepared similar LD-pruned sets of all SNPs significantly associated with the phenotype for each of the five traits…” would be more clear if it included something such as “...sets of all SNPs significantly associated with the phenotype, /irregardless of sex differentiation or not/,...”. Right now it’s unclear if this comparison set of SNPs includes those that are sex-diff or not. It becomes a bit more clear later on through the analyses, but explicitly delineating this here might help.

3) There are a few sentences that are awkwardly constructed. I would recommend revising them.

a) Lines 75-77: “/Large ranges in the degree of body size and shape sexually differentiated traits/ are repeatedly observed among…” (unclear here)

b) Lines 101-104: “/With the recent availability of a greatly increased participant sample size, this analysis represents a powerful extension of the several previous GWAS-based approaches/ that have studied the genetic architecture of…’ (feels clunky)

c) Lines 278-281: “...for the corresponding pruned set of all phenotype-associated SNPs (Figure 3B; Table S6), /using a permutation analysis/” (maybe start the sentence with this?)

4) This may not be necessary for the paper, but could the authors explain what the concern with BMI is? The paper cited has this at the end of its conclusion: “ABSI expresses the excess risk from high WC in a convenient form that is complementary to BMI and to other known risk factors.” Is this mostly a reflection of body shape being a better complementary phenotype to traits such as waist and hip circumference, whereas BMI may be too correlated? If so, the authors may want to be more explicit with their reasoning beyond ‘concerns about this metric’ -- this makes it seem like BMI is a potentially ‘bad metric’, which is maybe not the intention here.

**Have all data underlying the figures and results presented in the manuscript been provided?**

Reviewer #1: Yes

Reviewer #2: Yes

PLOS authors have the option to publish the peer review history of their article (what does this mean?). If published, this will include your full peer review and any attached files.

Reviewer #1: No

Reviewer #2: No

---

## [Decision Letter · Decision Letter 1]

20 Apr 2021

Dear Dr Arner,

We are pleased to inform you that your manuscript entitled "Patterns of recent natural selection on genetic loci associated with sexually differentiated human body size and shape phenotypes" has been editorially accepted for publication in PLOS Genetics. Congratulations!

Yours sincerely,

Scott M. Williams

Section Editor: Natural Variation

PLOS Genetics

Hua Tang

Section Editor: Natural Variation

PLOS Genetics

Comments from the reviewers (if applicable):

Reviewer's Responses to Questions

**Comments to the Authors:**

Reviewer #1: The authors were very responsive to my previous comments and have adequately addressed them. I think this is a really interesting and important paper and the analyses are well-done.

I just have a couple of minor comments to help improve the presentation.

1) Lines 146-149: This part is still a little confusing and may be missing some words. Or, maybe change “calculated” to “compared”.

2) Line 231: Thank you for the clarification in the response to reviewers about the rationale for comparing the trait effect size estimates between males and females. However, the part about “lower trait value variance” is still a little unclear. What is the variance of? The effect sizes? The traits? The difference in effect sizes? A few more words of clarification would go a long way here.

3) The authors might consider using a subscript on their t-sexDiff statistic. The dash could look like a minus sign and lead to confusion. For example, line 402, it could look like “t minus SexDiff=”.

Reviewer #2: Thank you to the authors for their responses to my previous suggestions and for the edits made to the manuscript. In particular I think the inclusion of the iHS and the GWAS catalog analyses provide additional, interesting perspectives to the results. I believe the authors have addressed my concerns adequately and that the manuscript has become stronger as a result of their additional work. As I was already mostly satisfied with the manuscript before, I feel that it is suitable for publication now as well. I have no additional comments.

**Have all data underlying the figures and results presented in the manuscript been provided?**

Reviewer #1: Yes

Reviewer #2: Yes

PLOS authors have the option to publish the peer review history of their article (what does this mean?). If published, this will include your full peer review and any attached files.

Reviewer #1: No

Reviewer #2: No

**Data Deposition**

http://datadryad.org/submit?journalID=pgenetics&manu=PGENETICS-D-20-01472R1

**Press Queries**

---

## [Editor Report · Acceptance letter]

12 May 2021

PGENETICS-D-20-01472R1 

Patterns of recent natural selection on genetic loci associated with sexually differentiated human body size and shape phenotypes 

Dear Dr Arner, 

We are pleased to inform you that your manuscript entitled "Patterns of recent natural selection on genetic loci associated with sexually differentiated human body size and shape phenotypes" has been formally accepted for publication in PLOS Genetics! Your manuscript is now with our production department and you will be notified of the publication date in due course.

With kind regards,

Katalin Szabo

PLOS Genetics

On behalf of:
